# Development of Computational Approaches with a Fragment-Based Drug Design Strategy: In Silico Hsp90 Inhibitors Discovery

**DOI:** 10.3390/ijms222413226

**Published:** 2021-12-08

**Authors:** Roberto León, Jorge Soto-Delgado, Elizabeth Montero, Matías Vargas

**Affiliations:** 1Facultad de Ingeniería, Universidad Andres Bello, Viña del Mar 2531015, Chile; elizabeth.montero@unab.cl (E.M.); m.vargasmarn@uandresbello.edu (M.V.); 2Departamento de Ciencias Químicas, Facultad de Ciencias Exactas, Universidad Andres Bello, Viña del Mar 2531015, Chile

**Keywords:** computational approach, fragment-based drug design, Hsp90, protein inhibitors

## Abstract

A semi-exhaustive approach and a heuristic search algorithm use a fragment-based drug design (FBDD) strategy for designing new inhibitors in an in silico process. A deconstruction reconstruction process uses a set of known Hsp90 ligands for generating new ones. The deconstruction process consists of cutting off a known ligand in fragments. The reconstruction process consists of coupling fragments to develop a new set of ligands. For evaluating the approaches, we compare the binding energy of the new ligands with the known ligands.

## 1. Introduction

Rational drug design remains a challenging problem for computational methods developed in bioinformatics and medicinal chemistry. In this sense, computational generative methods have begun to show promising results for the design problem. Principally, computer-aided drug discovery/design (CADD) has played a significant role in developing therapeutically important small molecules [1]. In general, the computational methods used in research of drug development are broadly classified into two groups: (i) ligand-based methods or (ii) structure-based methods [2].

The ligand-based methods use only ligand information for predicting activity depending on its similarity/dissimilarity to previously known active ligands. When the structural information of the protein is sufficient, it uses structure-based approaches. In other words, it is a simulation of the protein-ligand interactions. These interactions are essential for biochemical functionality and are present in all biochemical roles [3]. According to the structure-based technique, the development of a new compound consists of two stages [4]. The first one consists of the exploration of an ample conformational space representing various potential binding modes, and the second one of obtaining an accurate prediction of the interaction energy associated with each of the predicted binding conformations [5]. In this context, the molecular docking programs perform these tasks through a cyclical process, in which specific scoring functions evaluate the ligand conformation. This process is carried out recursively until converging to a minimum energy solution [5,6,7].

From the point of view of searching for active compounds on one or more therapeutic targets using compound libraries, virtual screening (VS) based on molecular docking is an efficient method to recover new successful compounds in drug discovery. On the other hand, fragment-based drug design (FBDD) has emerged as a robust approach to identify small molecules that bind to a wide range of therapeutic targets; this approach has led to efficient progress in developing new drugs [8,9,10,11,12]. Experimental and computational FBDD has established itself as an effective alternative to more traditional methods such as high-throughput screening (HTS) [13,14]. However, in computational FBDD, they have not yet utilized the power of three-dimensional (3D) structural information for this reason. Successful fragment making often requires co-structures of the fragments bound to their target proteins and various biophysical and biochemical assays to track potency and efficacy. We have incorporated molecular docking to measure the affinity for new ligands obtained from our method.

Additionally, the binding energy of the entire molecule with the target can be considered the sum of individual binding energy between the fragments and the target. Despite this affirmation, identifying suitable fragments that bind to the neighboring binding sites is an obstacle. An essential advantage of the FBDD approach is that once the detailed interaction within the pocket is experimentally validated and clearly understood, it could provide a huge opportunity to design potent drug-like compounds [12]. The construction of a fragment library is the first step for FBDD. Several factors should be considered, including [12]: (i) the difference between fragments and hits or leads, wherein [15] propose a rule-of-three (RO3) for the construction of a fragment (molecular weight is <300, cLogP is ≤3); (ii) the size of the fragment library; (iii) structural diversity of the fragment library; (iv) the solubility of fragments; and (v) the drug-likeness of fragments. We used the FBDD strategy called the deconstruction-reconstruction approach (DRA) in this work, which has gained attention in recent years [16]. The first step of DRA is to deconstruct known ligands into several fragments obtaining a fragment library. A size of 500 to 3000 components is recommended [17]. The second step is to reconstruct these fragments selected from the relevant fragment library into a new scaffold. Classic guidelines should govern the reconstruction stage, such as Lipinski’s rule of five and Veber’s rules to maintain suitable drug-like physicochemical properties, including molecular weight and volume [18,19]. For many years the DRA has been explored, showing it being helpful for target proteins. The main reason is that many reported ligands can provide robust data to create novel and promising scaffolds for further development.

This work presents two computational approaches aimed to decrease the searching time for potential high-quality ligands for Hsp90 inhibitors. The two approaches use a fragment-based drug design (FBDD) strategy for designing new ligands. A semi-exhaustive search strategy and a heuristic search algorithm that focuses on decreasing the computational required by the first approach. We have constructed a library of several Hsp90 ligands that have already been addressed [20,21,22,23,24] to evaluate our method. However, the problem of finding acceptable ligands with a high inhibition level remains unsolved. Currently, a database of more than 70 Hsp90 ligands [25] is available. We use this database for the experiments.

## 2. Materials and Methods

### 2.1. Semi-Exhaustive Approach

The semi-exhaustive approach follows an FBDD technique. Mainly, we use a DRA for the computations of new ligands. In Figure 1, we can observe the stages of the algorithm. In the following, we explain each step:

**Stage 1. Deconstruction.** A set of $N_{KL}$ Hsp90 ligands, called known ligands (KL), is selected for the fragmentation stage. Each ligand is cut off in sub-ligands considering the following constraints extracted from RO3 [15]: (i) the molecular weight must be <300; (ii) cLogP must be ≤3; (iii) the number of hydrogen bond donors must be ≤3 and (iv) the number of hydrogen bond receptors must be ≤3 [26]. Let $N_{KL}$ be the number of known ligands. Each ligand $L_{i},\forall i=1,\ldots ,N_{KL}$ is fragmented in $m_{i}$ sub-ligands (fragments). Then a total of $N_{FL}=\sum_{i=1}^{N}m_{i}$ fragments are generated from KL. Then the number of sub-ligands stored in the folder FL is $N_{FL}$.

**Stage 2. Duplicity and Cleaning.** For each sub-ligand obtained in the first stage, $N_{FL}$, we apply a duplicity check process. The process compares each sub-ligand with the rest of sub-ligands in the set to check if a duplication exists. For this goal, we use the Tanimoto coefficient, where the similarity between each pair of sub-ligands is computed [27]. If there are duplicated sub-ligands, they are deleted. Let $D_{F}$ be the number of duplicated sub-ligands, therefore the number of sub-ligands now is $N_{SL}=N_{FL}-D_{F}$. It is important to note that if a sub-ligand appears several times, all their copies are deleted, and just one remains.

**Stage 3. Sub-Ligands.** The folder SL* contains all sub-ligands obtained in the previous stage. Each sub-ligand that includes an asterisk identifier indicates that it contains a broken bond. Another folder called SL contains each sub-ligand without the asterisk identifiers. In this set, we have the same sub-ligands from SL*, but without the identifiers. We recall that each of these folders contain $N_{SL}$ sub-ligands.

**Stage 4. Reconstruction.** The folder SL* contains the header of each sub-ligand, i.e., the sub-ligand with the asterisks identifying broken bonds. The folder SL contains the body of the sub-ligands, i.e., the sub-ligands without the asterisk identifiers.

The process of union between sub-ligands starts taking a header from set SL* and a body from set SL. Let consider two sub-ligands, sub-ligand *A* (from SL*) and *B* (from SL).

The body of sub-ligand *B* bonds into the first asterisk identifier on the header of sub-ligand *A*. Then, if asterisk identifiers remain in the sub-ligand *A*, a random sub-ligand from SL is coupled. This last step is repeated until no free identifiers remain in *A*. The process is repeated for each sub-ligand *B* in SL with each sub-ligand *A* in SL*. Each new ligand obtained from the union process is stored in the folder NL. Let recall that we have $N_{SL}$ sub-ligands in each folder, SL* and SL, then the reconstruction process will generate $N_{SL}^{2}$ new ligands. If a new ligand does not satisfy Lipinski’s Rule of Five [18] and Veber’s rules [19], then it is rejected. Let $R_{NL}$ be the number of new ligands which are rejected, then the total of new ligands is $N_{NL}=N_{SL}^{2}-R_{NL}$.

**Stage 5. Duplicity and Cleaning.** As in Stage 2, a duplicity and cleaning process is applied in the same way. Each new ligand is compared with the rest of the ligands in set NL, checking for duplication. If so, then all the duplicated new ligands are deleted, and just one remains. Let $D_{NL}$ be the number of duplicated new ligands, then the number of new ligands now is $N=N_{NL}-D_{NL}$. Therefore the total number of new ligands obtained from known ligands is:(1)$$N={\left(N_{FL}-D_{F}\right)}^{2}-R_{NL}-D_{NL}$$
where $N_{FL}$ is the total fragments from KL library, $D_{F}$ is the number of deleted fragments due to duplicity, $D_{NL}$ is the number of new ligands deleted from duplicity, and $R_{NL}$ is the number of rejected new ligands.

**Stage 6. Validation.** The validation stage consists of computing the binding energy (BE) of each new ligand stored in NL. The process consists in calculating the interaction between the Hsp90 protein and a specific ligand. Then, considering several possible docking positions, the BE is computed. If the size of a ligand is larger than the grid specified for docking with the Hsp90, the evaluation is not possible. These ligands remain valid but not available. From all docking positions, it considers the most negative energy. The molecular docking software Autodock [28] is used in this stage.

**Deconstruction example.** The ligand PU3 is presented in Figure 2. If this ligand is cut off (deconstruction), then a total of five sub-ligands can be obtained. In Figure 2 the corresponding sub-ligands are presented as well.

The open-source software OpenBabel [29] is used to convert a ligand, from PDB (protein data bank) format to SMILES (simplified molecular-input line-entry system) format [30]. The line notation of SMILES format allows to work the molecular representation of the ligands in a simpler manner. The reading of files with chemical structures in this format becomes easier to handle than with PDB format. We use the SMILES format from stages 1 to 5 (deconstruction, clean/duplicity, sub-ligands/reconstruction, and clean/duplicity). The open-source software PyMol [31] is used for the visualization.

Automated docking was used to locate new compound appropriate binding orientations and conformations into the Hsp90 binding pocket. The structure of Hsp90 used in molecular docking was obtained from X-ray crystal data in RCSB Protein Data Bank (PDB ID: 5LQ9). The molecular docking calculations were carried out by using the AutoDock4 software. The protein structure was prepared by removing water, adding Kollman charges and polar hydrogen. First, the residues around 6Å of the co-crystallized ligand were considered the binding site for docking calculations. Note that this box centered in the co-crystallized ligand involved two characteristic regions: (i) Site 1, corresponding to ATP-binding site and (ii) Site 2, which represents an internal region (see Figure 3). All the molecules from the data set were docked into the active site of Hsp90. Then, a grid map was generated by the auxiliary program AutoGrid4 using x, y, and z coordinate to find a small but more representative grid center for the active site. The grid dimensions were set to 45×45×45 points, with a grid spacing of 0.375Å. The number of docking runs was set to 100. The population in the genetic algorithm was 150. After docking, the 100 docked poses were clustered into groups with RMS deviations lower than 1Å. Among the entire cluster of complexes predicted by AutoDock4, the most populated cluster conformation, together with the lowest energy conformation for the most active compound docked to the receptor.

The computation equipment used in the experiments is an Intel Core i7 with 3.2 GHz of processing on Ubuntu 16.04.

### 2.2. Heuristic Approach

We propose a heuristic-based approach to find new high-quality ligands from the sub-ligands obtained from the deconstruction process shown in Figure 1 in reduced computation times. This process works as a replacement for the reconstruction process shown in stage 4 in Figure 1.

The objective is to reduce the time involved in the reconstruction process using a local search approach capable of quickly constructing high-quality new ligands. Next, we present the representation and evaluation function used in our algorithm. Also, it shows the main structure of the local search approach and the search operators used.

#### 2.2.1. Representation

A list of fragments represents a solution. Each solution contains a unique header and one or more bodies. The number of bodies depends on the number of available bonds (asterisk identifiers) in the header structure.

Figure 4 shows an example of the representation of a solution. The solution contains the header in the first position, followed by two bodies. It uses SMILES format for the elements in the list. In the lower area, the two-dimensional model is shown as well.

#### 2.2.2. Evaluation Function

The evaluation starts checking the feasibility of the list of fragments and then evaluates its quality. Feasibility is a filter according to the Lipinski’s rules to determine that each new ligand is valid in a pharmacological environment [18]. Once validated, it computes the energy of the new ligand using the AutoDock Vina [32] software and the same conditions described in Section 2.1. The number of positions to test is considered a parameter of the algorithm (pos).

#### 2.2.3. General Structure

Algorithm 1 shows the structure of the local search algorithm proposed here. It corresponds to a simulated annealing-based approach that uses two repairing operators to generate a new ligand at each iteration. It can accept worse quality solutions based on a probability controlled by a cooling schedule. This cooling schedule allows more deteriorations in the early stages of the search process and reduces their acceptance at the end of the process. The algorithm requires the parameter values for reduction factor (α) and initial and final temperature (initial_temp and final_temp).

The algorithm performs a fixed number of times the whole simulated annealing procedure. This number of repetitions is required to change the header structure of the current solution during the process. The parameter max_tries controls the number of repetitions. In each step, a local search process iterates from the given initial temperature to the final temperature (lines 5 to 22). At each iteration, temperature is reduced by α factor (line 21).
**Algorithm 1** Simulated Annealing Algorithm  1:BestLigand ←∅;  2:**while** (tries < max_tries) **do**  3: current_temp ← initial_temp;  4: CurrentLigand ← generate_initial_ligand();  5: **while** (current_temp > final_temp **and not** stuck) **do**  6:   NewLigand ← Change (CurrentLigand);  7:   **if** (*f*(NewLigand) ≤*f*(CurrentLigand)) **then**  8:    CurrentLigand ← NewLigand;  9:    NewLigand ← Swap (CurrentLigand);10:    **if** (*f*(NewLigand) ≤*f*(CurrentLigand)) **then**11:    CurrentLigand ← NewLigand;12:    **end if**13:   **else**14:    **if** (rand (0,1)<exp((f(NewLigand)−f(CurrentLigand))/current_temp)) **then**15:    CurrentLigand ← NewLigand;16:    **end if**17:   **end if**18:   **if** (*f*(CurrentLigand) ≤*f*(BestLigand)) **then**19:    BestLigand ← CurrentLigand;20:   **end if**21:   current_temp ←α· current_temp;22: **end while**23:**end while**

Each simulated annealing process incorporates a stuck criterion to reduce the computational effort dedicated to exploring local search neighborhoods. The parameter stuck controls the number of iterations without improvement. Variable BestLigand stores the best solution found during the whole process (line 1). Simulated annealing starts each iteration from a new random solution (line 4). To select a high-quality initial solution, a competition between random ligands is performed. The description of this procedure is presented in Section 2.2.4.

The approach implements two movements: *change* and *swap*. At each iteration, the process searches for the best combination of bodies for a given header. Hence, the first movement selects a random body from the solution and changes it by a different body available in the library (line 6). If the movement generates an improvement in the energy of the current ligand, a second movement is applied to evaluate the distribution of these bodies in the available bonds of the header (line 9). Movements are explained in Section 2.2.5 and Section 2.2.6.

The algorithm performs the simulated annealing cooling scheme to accept worsening solutions only during the use of the change operator (line 14). It is expected that this operator generates more significant changes in the structure of solutions, and hence, it is allowed to generate deteriorations in some possible ligands as well.

#### 2.2.4. Initial Solution Generation

In this step, the algorithm performs a competition between random solutions. First, it generates a solution choosing its header structure from the available ones. From its available $n_{b}$ bonds, the algorithm selects $n_{b}$ bodies to complete a new ligand. The selection of these bodies is random.

#### 2.2.5. Change Movement

This movement consists of changing one of the bodies in the current solution. A random body from the current solution is selected and replaced by a new random body. To reduce the neighborhood of this movement, we use the Tanimoto indicator to determine the similarity between bodies. A Tanimoto similarity higher than 0.0 defines that two bodies belong to the same neighborhood.

Figure 5 shows an example of the application of this movement. An input ligand and its corresponding output are in the upper and lower part, respectively. Each one is shown with its SMILES format and its two-dimensional representation. Moreover, it also shows the two-dimensional structure of both corresponding ligands on the right of the figure. In this example, the first body is replaced by a new body randomly selected using its Tanimoto similarity neighborhood from the SL set.

#### 2.2.6. Swap Movement

This movement changes the order of the locations of bodies in the available header’s bonds. This step occurs after applying the change operator to the bodies. The goal of this movement is to improve the quality of the solution only if a promising new ligand was found. It is important to remark that the movement is not applied if the current solution contains only one body. Figure 6 shows an example of the use of this movement. It shows the same information as in Figure 5. This input ligand corresponds to the output ligand obtained in the change movement example. The movement only changes the location of two bodies, but this can lead to essential changes in the final ligand structure. It shows these changes in the right part of Figure 6 with the two-dimensional representation of the resulting ligands.

## 3. Results

An Hsp90 ligand library with 70 components is selected and used [25]. As a preliminary check, we evaluate the binding energy (BE) for each ligand. The BE is the best binding energy from 25 docking positions. The worst energy level found is 9.2, and the best one is −11.3. Two ligands (a hydroxy-indazole and an amino-quinazoline) reach the best value. The average BE is −9.2, and the 92% of ligands in the library have energy levels lower than −6.9 and closer to 40% lower than −10.

The experiment design is the following: (i) it considers eight groups of ligands (eight different KL libraries), where each one has a different number of ligands, (ii) for each group, the semi-exhaustive process obtains a new ligand library, (iii) for each group, the heuristic approach obtains a new ligand library, and (iv) the BE is computed for each new ligand from both libraries.

Each ligand from the KL library belongs to a particular family: (A) resorcinol, (B) hydroxy-indazole, and (C) “others” (including aminothienopyridine, 6-hydroxyindole, 7-imidazopyridine, 2-aminopyridine, adenine, 7-azaindole, aminopyrrolopyrimidine, benzamide, and aminoquinazoline). For more details see Table 1, Table 2 and Table 3.

The criterion for defining the groups is to keep diversity between the ligands, except group 8, where the selection is from the two main families (resorcinol and hydroxy-indazole). For instance, group 1 contains a ligand from the resorcinol family, a ligand from the hydroxy-indazole family, and three ligands from others. As the number of ligands increases in the groups, the diversity remains. The main goal of the groups definition is that every ligand belongs at least to one group. For each group, the number of ligands from the resorcinol and hydroxy-indazole families together is similar to the number of ligands from “others”. Group 8 is the exception, where there are no ligands from “others”. Figure 7 shows the Tanimoto matrix indicating the similarity level between ligands in the library grouped by families. We can observe that there is a high similarity between ligands from the same family. A low similarity between ligands from different families can be observed as well. As mentioned before, the incorporation of each ligand in a group is focused in keeping a high diversity in terms of similarity.

Table 4 presents the eight groups representing the KL libraries for the experiments.

Table 5 shows the results of the semi-exhaustive approach performance. It shows the number of known ligands (KL), the best binding energy obtained from KL, the number of fragments (sub-ligands) generated, the number of new ligands (NL), the best binding energy from NL, and the total computational time for each group.

Interestingly, the semi-exhaustive approach always obtained new ligands that show better binding energy than the best in the corresponding KL library. We can confirm that the deconstruction-reconstruction method can generate high-quality new ligands for the Hsp90. The semi-exhaustive approach obtains a minimum improvement of around 1% for group 8 and a maximum of 40% for group 1. Moreover, it obtains an average improvement of around 16% on binding energy.

From Table 5 it is also interesting to notice that the number of new ligands and the execution time do not necessarily grow exponentially according to the size of the KL library of each group. As expected, this is due to the semi-exhaustive nature of the approach and its duplicity and cleaning stages. Group 1 required only 89.1 min to obtain a total of 581 new ligands reconstructed from a library of 5 known ligands. Meanwhile group 8 required around to 3660 min to obtain 51,514 new ligands from a group of 40 known ligands.

The execution time increases with the number of new ligands, but the binding energy does not show the same behavior. Probably, it is because the process can reconstruct new high-quality ligands from fragments available in ligands belonging to different families, but not necessarily from a large KL library.

Figure 8 shows the structure of the best new ligand obtained by the semi-exhaustive approach in each group. For simplicity, Table 6 shows each structure in the SMILES format.

Table 7 summarizes the results obtained by the heuristic search approach. It shows the average best binding energy, the standard deviation, and the best binding energy obtained from five executions using different random seeds. It also shows the average computational time and standard deviation of these executions.

Interestingly, the best binding energy obtained by the heuristic algorithm is always close to the results obtained by the semi-exhaustive approach.

The heuristic approach shows a stable behavior in terms of execution time. For group 8, five executions can take around 200 min, 6% of the time required by the semi-exhaustive approach. An advantage of the heuristic approach is being able to work with large KL libraries. This method can be used as an explorative supporting system before the application of the semi-exhaustive or exhaustive version of the first approach.

Figure 9 shows the structure of the best new ligand obtained with the heuristic approach in each group. For simplicity, Table 8 shows each new ligand structure in the SMILES format. Figure 10 shows the eight best docking position obtained for each group by the semi-exhaustive approach and the heuristic search approach.

## 4. Discussions

According to the results, the best binding energy from each group shows an improvement compared to the best energy from the known ligands library (−11.3). Group 8 obtained the worst improvement, while groups 4 and 7 got the highest improvements using the semi-exhaustive approach. We can hypothesize that the variety of the group in terms of the families to which the known ligands belong can influence the binding energy improvement. We can observe that the binding energies obtained in the experiments of group 8 are of lower quality compared to groups 4 and 7. Group 8 is composed exclusively by ligands from the two leading families, while groups 4 and 7 consider ligands from all families. Then, a high variety of fragments and new ligands can be expected during the deconstruction and reconstruction processes. Group 1 obtained the best improvement with an increment of 40% between the best binding energy of the known ligands and the new ligands. Group 8 obtained the worst improvement with just 1%. According to the new ligands obtained with the semi-exhaustive approach, groups 4 and 6 got the same new ligand. However, the binding energy is quite different because the computation of the binding energy uses an approximation method. The final value is not necessarily the same but close. All ligands obtained with the semi-exhaustive approach are structurally similar. A 67% presents slight structural variations.

The heuristic approach obtains high-quality results with low computational times. This approach can be used for studying more extensive and diverse KL libraries. If the new ligands have promising results, then it can be explored exhaustively with the first approach.

According to the computational time, the trend is linear, as we can observe in Figure 11 with the semi-exhaustive approach. However, the computational time with the heuristic search is lower. This result is significant because it allows us to explore a higher number of known ligands.

It is essential to notice that the presented approaches are computational strategies that aim to decrease the search time of new high-quality ligands. Hence, only computational tools have validated the results obtained. A biological evaluation will be part of the future work.

## 5. Conclusions

This work presents a semi-exhaustive approach and a heuristic search algorithm based on the deconstruction-reconstruction method for drug design for the Hsp90 inhibition. The results are promising, looking at a future work where we can explore larger set of ligands. The critical point is that the reconstruction algorithm is a stage where we can use several computational techniques, generating a high level of diversity of new ligands. An important challenge is that the problem can scale quickly. This could be addressed using multi-core programming or GPU computing.

The focus explored in this work is obtaining a more significant number of ligands and getting a higher quality of the new ligands by using combinatorial methods. Another critical point is to propose an improved reconstruction process that generates even more diverse molecular structures than the ones developed in this work. An intuitive idea leads us to think that ligands with various combinatorial structures can induce new ones with higher quality in terms of binding energy. This approach can scale to other aspects and other pathologies.

The deconstruction-reconstruction method allows generating high-quality new ligands based on fragments. Independently of the reconstruction technique, the results show that it is always possible to find promising new ligands. Furthermore, our experiments show that the binding energy obtained with the heuristic search is close to the semi-exhaustive approach. However, the computational time used with the heuristic approach is considerably lower than the semi-exhaustive. Then, we recommend the use of heuristic search techniques for finding promising new ligands. These techniques allow obtaining high-quality binding energy in acceptable computational times.

## Figures and Tables

**Figure 1 ijms-22-13226-f001:**
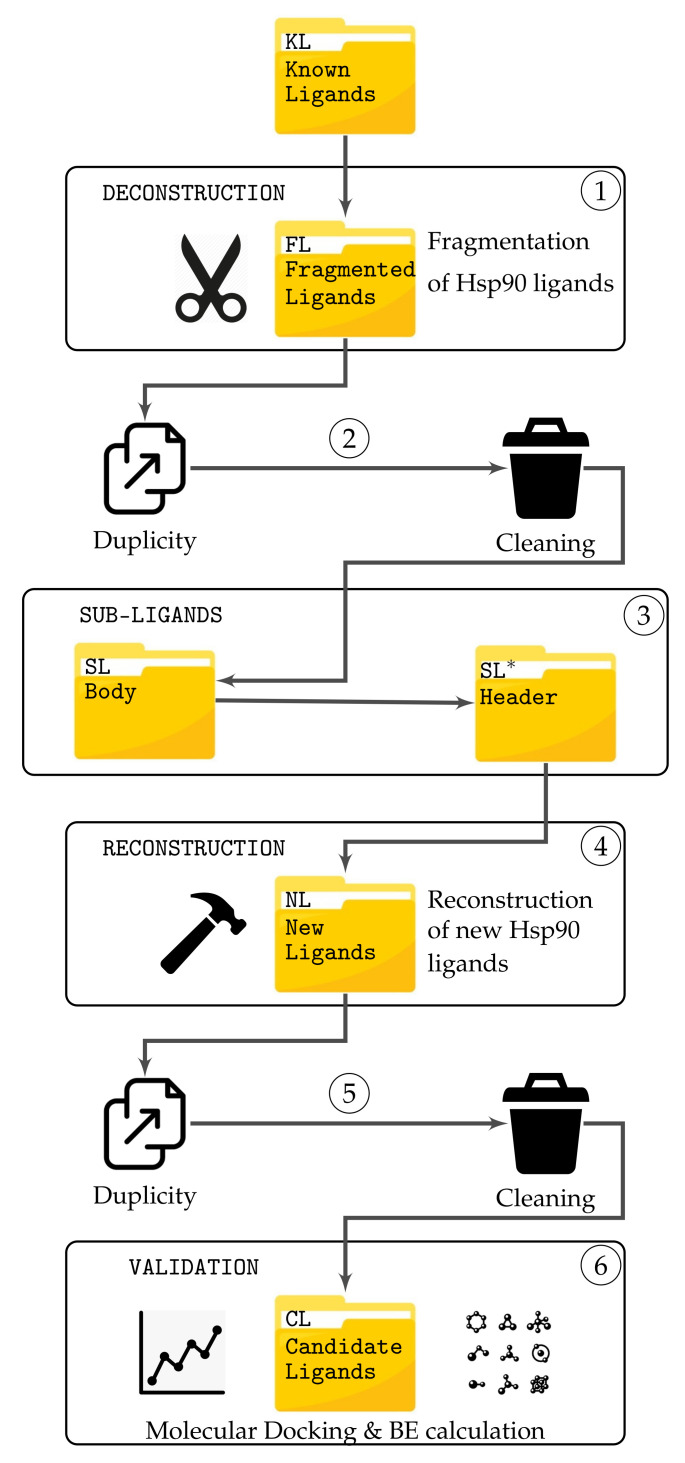
Flow chart of the deconstruction-reconstruction approach applied to HSP90 ligands.

**Figure 2 ijms-22-13226-f002:**
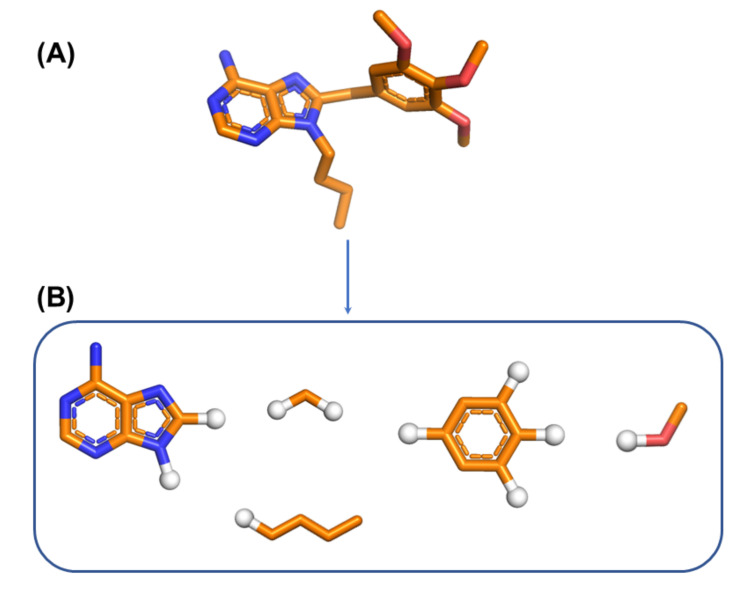
(**A**) PU3 ligand of Hsp90 and (**B**) The five sub-ligands from PU3.

**Figure 3 ijms-22-13226-f003:**
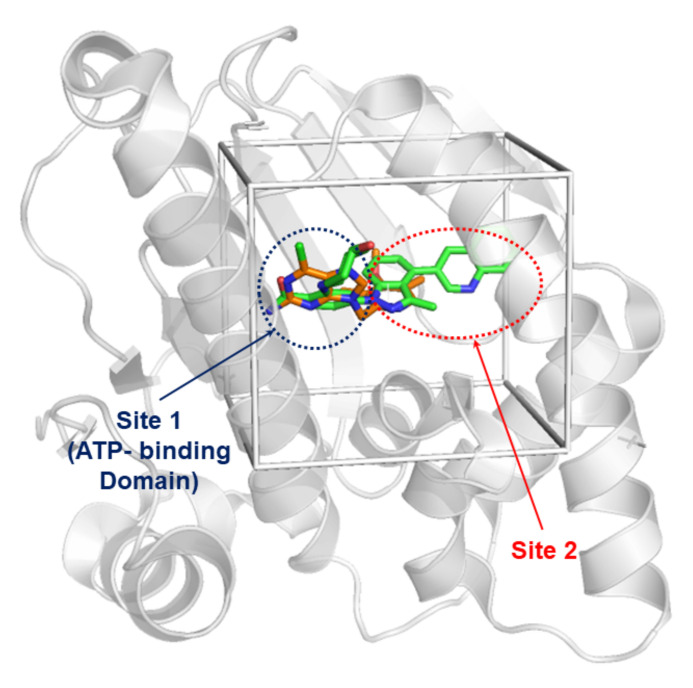
Hsp90 co-crystalized (PDB ID: 5LQ9) with ligand in green, and superposition of BIIB021 purine derivative compound in orange.

**Figure 4 ijms-22-13226-f004:**
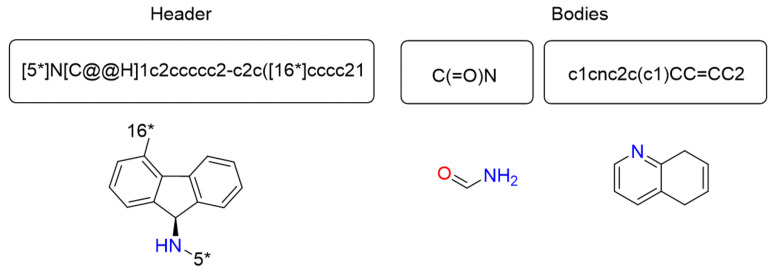
Representation example: A solution composed by one header and two bodies.

**Figure 5 ijms-22-13226-f005:**
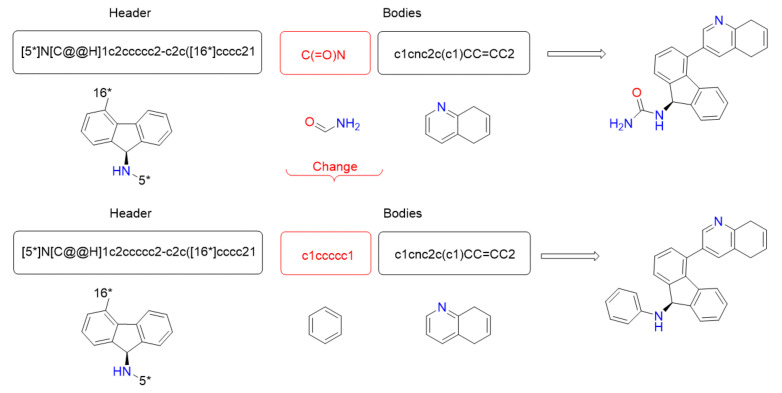
Change movement example.

**Figure 6 ijms-22-13226-f006:**
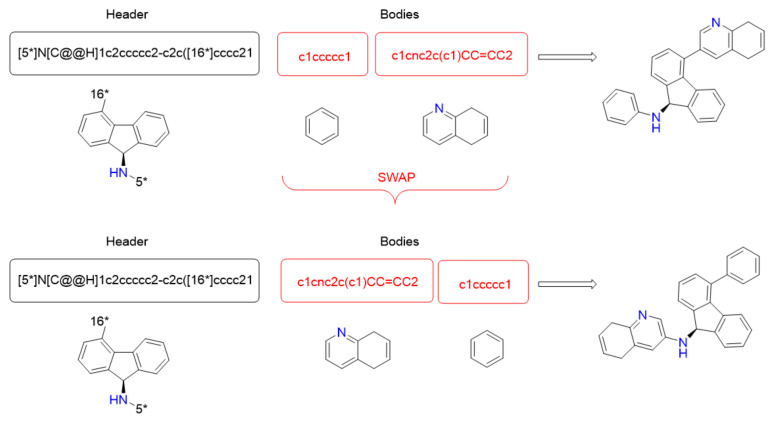
Swap movement example.

**Figure 7 ijms-22-13226-f007:**
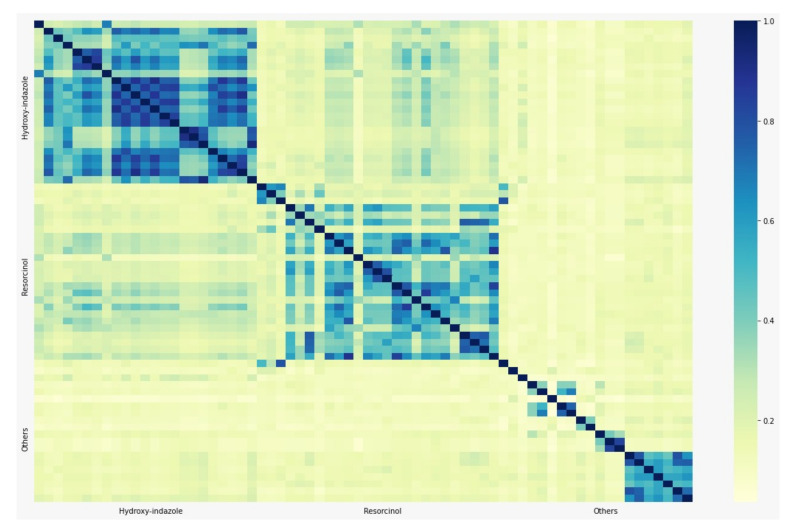
Tanimoto similarity matrix.

**Figure 8 ijms-22-13226-f008:**
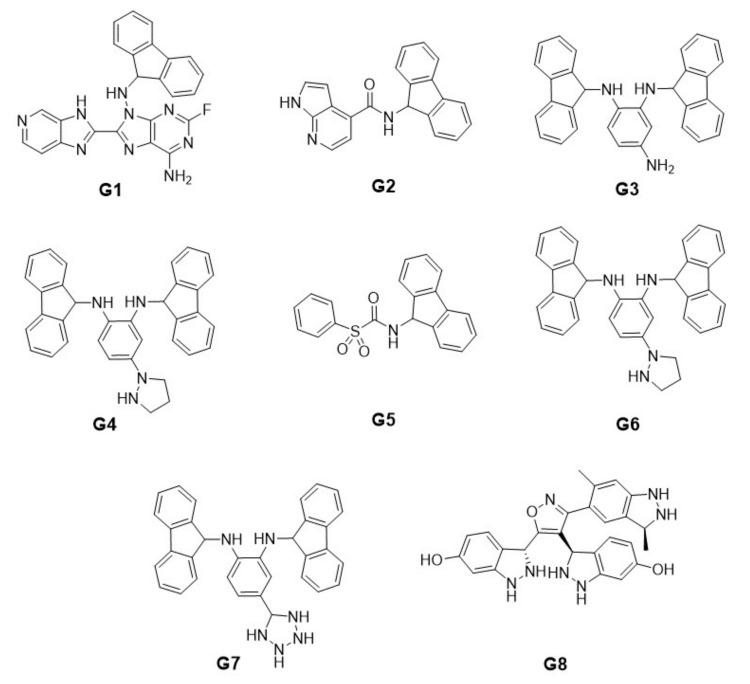
New ligand structures with the best binding energy obtained with the semi-exhaustive approach for each group.

**Figure 9 ijms-22-13226-f009:**
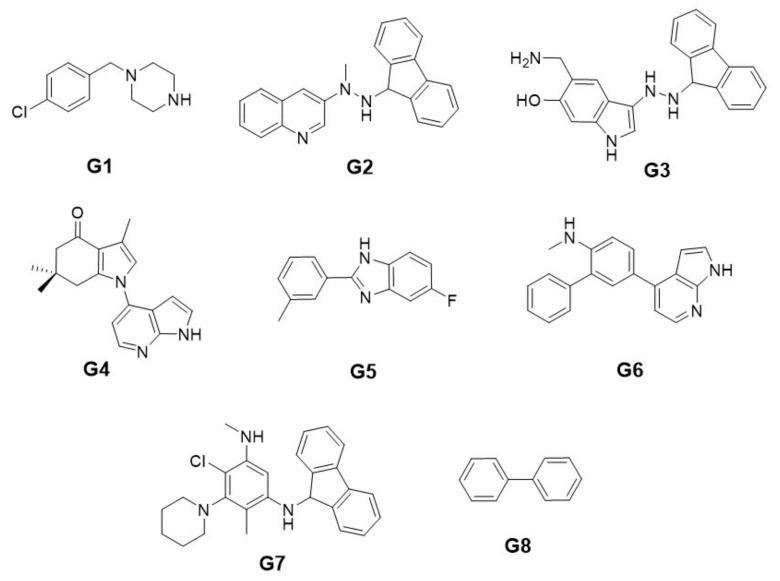
New ligand structures with the best binding energy obtained with the heuristic search approach for each group.

**Figure 10 ijms-22-13226-f010:**
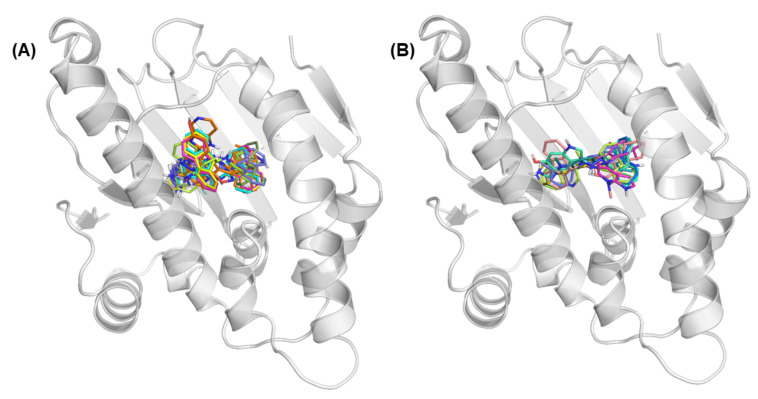
Best docking position for each new ligand obtained by (**A**) semi-exhaustive approach and (**B**) heuristic search approach.

**Figure 11 ijms-22-13226-f011:**
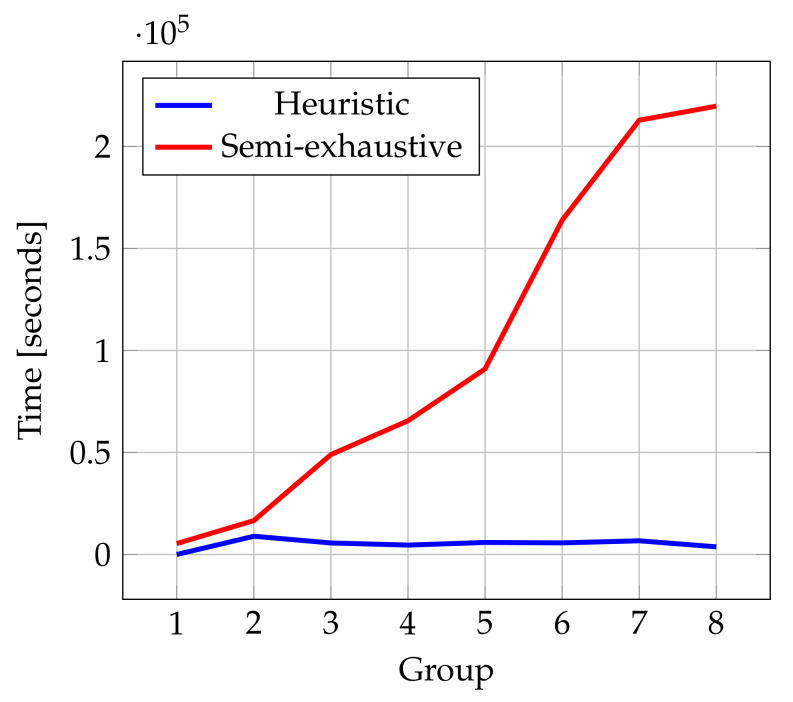
Computational time for each group.

**Table 1 ijms-22-13226-t001:** Hsp90 inhibitors: resorcinol [25].

Id	SMILES	BE
M01	CCNC(=O)c1noc(-c2cc(Cl)c(O)cc2O)c1-c1ccc(OC)cc1	-7.9
M02	CCNC(=O)c1[nH]nc(-c2cc(Cl)c(O)cc2O)c1-c1ccc(OC)cc1	-8.0
M03	CCNC(=O)c1noc(c2cc(Cl)c(O)cc2O)c1c3ccc(C[NH+]4CCOCC4)cc3	-7.6
M04	O=c1[nH]nc(-c2cc(Br)c(O)cc2O)n1-c1ccccc1F	-8.1
M06	CCc1cc(-c2n[nH]c(C)c2-c2ccccc2F)c(O)cc1O	-8.7
M07	O=c1[nH]nc(-c2ccc(O)cc2O)n1-c1ccccc1F	-8.3
M09	COc1ccc(-c2c(-c3ccc(O)cc3O)n[nH]c2C)cc1	-8.3
M10	CN(Cc1ccco1)C(=O)c1cc(-c2n[nH]c(=O)n2-c2ccccc2F)c(O)cc1O	-9.6
M11	Cc1ccccc1-n1c(-c2cc(C(=O)N(C)Cc3cccs3)c(O)cc2O)n[nH]c1=O	-8.9
M12	CCCCN(C)C(=O)c1cc(-c2n[nH]c(=O)n2-c2ccccc2F)c(O)cc1O	-9.1
M13	Oc1cc(O)c(-c2ccnn2-c2ccccc2Cl)cc1CCc1ccccn1	-9.7
M14	CCCN(C)S(=O)(=O)c1cc(-c2n[nH]c(=O)n2-c2ccccc2F)c(O)cc1O	-8.6
M15	CC(C)N(C)S(=O)(=O)c1cc(-c2n[nH]c(=O)n2-c2ccccc2F)c(O)cc1O	-8.9
M25	CC(C)N(C)S(=O)(=O)c1cc(-c2n[nH]c(=O)n2-c2ccccc2Cl)c(O)cc1O	-8.1
M26	CCCN(C)C(=O)c1cc(-c2n[nH]c(=O)n2-c2ccccc2C)c(O)cc1O	-8.4
M27	Cc1ccccc1-n1c(-c2cc(C(=O)N(C)Cc3ccccc3)c(O)cc2O)n[nH]c1=O	-9.3
M28	CCCCCCN(C)C(=O)c1cc(-c2ccnn2-c2ccccc2C)c(O)cc1O	-9.0
M29	Cc1cccc(CN(C)C(=O)c2cc(-c3n[nH]c(=O)n3-c3ccccc3C)c(O)cc2O)c1	-10.5
M30	Cc1ccccc1-n1c(-c2cc(C(=O)N(C)CC3CCCO3)c(O)cc2O)n[nH]c1=O	-8.7
M31	CCCCN(C)C(=O)c1cc(-c2n[nH]c(=O)n2-c2ccccc2)c(O)cc1O	-9.3
M32	Cc1ccccc1-n1nccc1-c1cc(C(=O)N(C)Cc2ccco2)c(O)cc1O	-9.2
M33	Cc1ccccc1-n1c(-c2ccc(O)cc2O)n[nH]c1=O	-8.2
M34	O=c1[nH]nc(-c2ccc(O)cc2O)n1-c1ccccc1Cl	-8.1
M35	CCc1ccccc1-n1c(-c2ccc(O)cc2O)n[nH]c1=O	-8.2
M36	CCCN(C)C(=O)c1cc(-c2n[nH]c(=O)n2-c2ccccc2F)c(O)cc1O	-9.0
M61	CCNC(=O)c1noc(-c2cc(C(C)C)c(O)cc2O)c1-c1ccc(C[NH+]2CCOCC2)cc1	-8.2

**Table 2 ijms-22-13226-t002:** Hsp90 inhibitors: hydroxy-indazole [25].

Id	SMILES	BE
M08	Cc1n[nH]c2cc(O)c(-c3ccnn3-c3ccccc3)cc12	-8.7
M37	Cc1cccc(Cc2n[nH]c3cc(O)c(C(=O)N(C)c4ccc5c(c4)OCO5)cc23)c1	-11.3
M38	C[NH+]1CCC(c2ccc(N(C)C(=O)c3cc4c(CCC(C)(C)C)n[nH]c4cc3O)cc2)CC1	-9.2
M39	CCCCN(C)C(=O)c1n[nH]c2cc(O)c(C(=O)N(C)c3ccc(N4CCOCC4)cc3)cc12	-9.0
M40	CN(Cc1ccc(Cl)cc1)C(=O)c2cc3c(Cc4ccccc4)n[nH]c3cc2O	-10.6
M41	Cc1cccc(Cc2n[nH]c3cc(O)c(C(=O)N(C)Cc4ccccc4)cc23)c1	-10.5
M42	Cc1cccc(Cc2n[nH]c3cc(O)c(C(=O)N(C)Cc4ccc(Cl)cc4)cc23)c1	-10.4
M43	Oc1cc2[nH]nc(Cc3ccccc3)c2cc1-c1ccnn1-c1ccccc1	-9.7
M44	Cc1ccc(N(C)C(=O)c2cc3c(Cc4cccc(C)c4)n[nH]c3cc2O)cc1	-10.8
M45	Cc1cccc(Cc2n[nH]c3cc(O)c(C(=O)N(C)c4ccc(N5CCOCC5)cc4)cc23)c1	-10.7
M46	Cc1cccc(Cc2n[nH]c3cc(O)c(C(=O)N(C)c4ccccc4)cc23)c1	-10.5
M47	Cc1cccc(Cc2n[nH]c3cc(O)c(C(=O)N(C)c4ccc(N5CCCCC5)cc4)cc23)c1	-10.9
M48	Cc1cccc(Cc2n[nH]c3cc(O)c(C(=O)N(C)c4ccc(N(C)C)cc4)cc23)c1	-10.4
M49	Cc1cccc(Cc2n[nH]c3cc(O)c(C(=O)N(C)c4ccc(N5CC[NH2+]CC5)cc4)cc23)c1	-10.7
M50	Cc1cccc(Cc2n[nH]c3cc(O)c(C(=O)N(C)c4ccc(N5CCN(C)CC5)cc4)cc23)c1	-10.1
M62	CO[C@H]1CCN(C(=O)c2n[nH]c3cc(O)c(C(=O)N(C)c4ccc(N5CCOCC5)cc4)cc23)C1	-8.4
M63	CO[C@H]1CCCN(C(=O)c2n[nH]c3cc(O)c(C(=O)N(C)c4ccc(N5CCOCC5)cc4)cc23)C1	-9.5
M64	CN(C(=O)c1cc2c(C(=O)N3CCCC3)n[nH]c2cc1O)c1ccc(N2CCOCC2)cc1	-9.9
M65	Cc1cccc(Cc2n[nH]c3cc(O)c(C(=O)N(C)c4ccc(N5CCOCC5=O)cc4)cc23)c1	-10.3
M66	Cc1cccc(Cc2n[nH]c3cc(O)c(C(=O)N(C)c4ccc(F)cc4)cc23)c1	-10.7
M67	COc1cccc(N(C)C(=O)c2cc3c(Cc4cccc(C)c4)n[nH]c3cc2O)c1	-10.8
M68	Cc1cccc(Cc2n[nH]c3cc(O)c(C(=O)N(C)c4cccc(C)c4)cc23)c1	-11.0
M69	CN(C(=O)c1cc2c(cc1O)[nH]nc2C(=O)N1CCOCC1)c1ccc(N2CCOCC2)cc1	-10.3

**Table 3 ijms-22-13226-t003:** Hsp90 inhibitors: “others” [25].

Id	SMILES	BE
M05	COc1ccc(-c2c(C#N)c(N)nc3sc(C(N)=O)c(N)c23)cc1OCCCC(=O)O	-8.4
M17	N#Cc1ccc(N2CCN(CCCc3c[nH]c4cc(O)c(C#N)cc34)CC2)cc1	-8.7
M18	Brc1cnc2[nH]cnc2c1C(=O)NC1c2ccccc2-c2c(-c3cnc4ccccc4c3)cccc21	4.7
M24	Nc1cc(C(=O)NC2c3ccccc3-c3c(-c4nc5ccncc5[nH]4)cccc32)ccn1	-4.5
M60	C#CCCCn1c(Cc2cc(OC)c(OC)c(OC)c2Cl)nc2c(N)nc(F)nc21	-8.0
M19	c1cnc2[nH]ccc2c1C(=O)N[C@@H]1c2ccccc2c2c1cccc2c1[nH]c2c(n1)cc(cc2)F	4.2
M20	O=C(NC1c2ccccc2-c2c(-c3nc4ccncc4[nH]3)cccc21)c1ccnc2[nH]ccc12	0.2
M58	Cc1cnc(Cn2ccc3c(Cl)nc(N)nc32)c(C)c1Cl	-9.0
M59	COc1c(C)cnc(Cn2cc(C#CCC(C)(C)O)c3c(Cl)nc(N)nc32)c1C	-6.9
M21	Cc1nn(-c2ccc(C(N)=O)c(N[C@H]3CC[C@H](O)CC3)c2)c2cccc(-c3cnc4ccccc4c3)c12	9.2
M22	Cc1cn(-c2ccc(C(N)=O)c(N[C@H]3CC[C@H](O)CC3)c2)c2c1C(=O)CC(C)(C)C2	-10.8
M23	Cc1cn(-c2ccc(C(N)=O)c(NC3CCC(=O)CC3)c2)c2c1C(=O)CC(C)(C)C2	-10.9
M51	Nc1nc(C(=O)N2Cc3ccc(O)cc3C2)c2ccccc2n1	-11.0
M52	Nc1nc(C(=O)N2Cc3ccccc3C2)c2cc(O)ccc2n1	-10.5
M53	C[NH+]1CCN(S(=O)(=O)c2ccccc2-c2ccc3nc(N)nc(C(=O)N4Cc5ccccc5C4)c3c2)CC1	-7.7
M54	CNCc1ccccc1-c1ccc2nc(N)nc(C(=O)N3Cc4ccccc4C3)c2c1	-11.3
M55	Nc1nc(C(=O)N2Cc3ccccc3C2)c2cc(-c3cc(F)c(F)cc3CCc3nnn[nH]3)ccc2n1	-10.8
M56	Nc1nc(C(=O)N2Cc3ccccc3C2)c2ccccc2n1	-11.0
M71	Cc1ccc2nc(N)nc(C(=O)N3Cc4ccccc4C3)c2c1	-11.0

**Table 4 ijms-22-13226-t004:** Groups for the experiment. Each ligand belongs to some family [25]. The ligand structure specification is in Table 1, Table 2 and Table 3. The **bold**, *cursive* or underlined ligand belongs to the **resorcinol**, *hydroxy-indazole* or “others” family respectively.

Group	KL Library
1	**M61**, *M08*, M17, M24, M60
2	**M35**, **M36**, *M69*, *M70*, M17, M24, M60, M05, M18, M19
3	**M01**, **M02**, **M03**, *M37*, *M38*, *M39*, M17, M24, M60, M05, M18, M19, M20, M21, M59
4	**M31**, **M32**, **M33**, **M34**, *M65*, *M66*, *M67*, *M68*, M17, M24, M60, M05, M18, M19, M20, M21, M22, M51, M58, M59
5	**M04**, **M06**, **M07**, **M09**, **M10**, *M40*, *M41*, *M42*, *M43*, *M44*, M17, M24, M60, M05, M18, M19, M20, M21, M22, M23, M51, M52, M53, M58, M59
6	**M25**, **M26**, **M27**, **M28**, **M29**, **M30**, *M48*, *M49*, *M50*, *M62*, *M63*, *M64*, M17, M24, M60, M05, M18, M19, M20, M21, M22, M23, M51, M52, M53, M71, M54, M55, M58, M59
7	**M11**, **M12**, **M13**, **M14**, **M15**, **M25**, **M26**, **M27**, *M45*, *M46*, *M47*, *M48*, *M49*, *M50*, *M62*, *M63*, M17, M24, M60, M05, M18, M19, M20, M21, M22, M23, M51, M52, M53, M71, M54, M55, M56, M58, M59
8	**M61**, **M01**, **M02**, **M03**, **M04**, **M06**, **M07**, **M09**, **M10**, **M11**, **M27**, **M28**, **M29**, **M30**, **M31**, **M32**, **M33**, **M34**, **M35**, **M36**, *M08*, *M37*, *M38*, *M39*, *M40*, *M41*, *M42*, *M43*, *M44*, *M45*, *M50*, *M62*, *M63*, *M64*, *M65*, *M66*, *M67*, *M68*, *M69*, *M70*

**Table 5 ijms-22-13226-t005:** The number of known ligands (# KL), the best binding energy (${\mathrm{BBE}}_{\mathrm{KL}}$) of known ligands, the number of fragments (# SL), the number of new ligands (# NL), the best binding energy (${\mathrm{BBE}}_{\mathrm{NL}}$) of new ligands and the computational time ($T_{S}$) in minutes for each group with the semi-exhaustive approach.

G	# KL	${\mathrm{BBE}}_{\mathrm{KL}}$	# SL	# NL	${\mathrm{BBE}}_{\mathrm{NL}}$	$T_{S}$
1	5	−8.7	25	581	−12.2	89.1
2	10	−10.6	48	2061	−12.8	276.8
3	15	−11.3	81	5848	−12.4	815.9
4	20	−11.0	95	7861	−13.0	1092.2
5	25	−11.0	114	12,266	−12.6	1515.7
6	30	−11.3	153	10,163	−12.9	2730.4
7	35	−11.3	177	28,910	−13.0	3546.9
8	40	−11.3	237	51,514	−11.4	3661.7

**Table 6 ijms-22-13226-t006:** New ligands with best binding energy using the semi-exhaustive approach.

G	New Ligand (SMILES)
1	[C@H]1(N[C@@H]2[C@@H](N)N=C(F)N[C@H]2N1NC1c2ccccc2c2ccccc12)
	[C@@H]1Nc2ccncc2N1
2	C(=O)(c1ccnc2[nH]ccc12)NC1c2ccccc2c2ccccc12
3	c1(ccc(N)cc1NC1c2ccccc2c2ccccc12)NC1c2ccccc2c2ccccc12
4	c1(ccc(cc1NC1c2ccccc2c2ccccc12)N1CCCN1)NC1c2ccccc2c2ccccc12
5	C(=O)([SH](O)(O)c1ccccc1)NC1c2ccccc2c2ccccc12
6	c1(ccc(cc1NC1c2ccccc2c2ccccc12)N1CCCN1)NC1c2ccccc2c2ccccc12
7	c1(ccc(cc1NC1c2ccccc2c2ccccc12)[C@@H]1NNNN1)NC1c2ccccc2c2ccccc12
8	[C@H]1(ONC(=C1[C@@H]1NNc2cc(O)ccc12)c1cc2[C@H](C)NNc2cc1O)
	[C@@H]1NNc2cc(O)ccc12

**Table 7 ijms-22-13226-t007:** The best binding energy (${\mathrm{BBE}}_{\mathrm{KL}}$) of known ligands, the best binding energy (${\mathrm{BBE}}_{\mathrm{NL}}^{S}$) and the computational time ($T_{S}$) in minutes with the semi-exhaustive approach, the average binding energy (${\mathrm{ABE}}_{\mathrm{NL}}$), the best binding energy (${\mathrm{BBE}}_{\mathrm{NL}}^{H}$) and the average computational time ($T_{H}$) in minutes for each group with the heuristic search approach. The binding energy equal or lower than the obtained by the semi-exhaustive approach is shown in **bold**.

G	${\mathrm{BBE}}_{\mathrm{KL}}$	${\mathrm{BBE}}_{\mathrm{NL}}^{S}$	$T_{S}$	${\mathrm{ABE}}_{\mathrm{NL}}\pm \sigma$	${\mathrm{BBE}}_{\mathrm{NL}}^{H}$	$T_{H}\pm \sigma$
1	−8.7	−12.2	89.1	−12.04 ± 0.3	**−12.4**	76.2 ± 20.2
2	−10.6	−12.8	276.8	−12.00 ± 0.1	−12.3	104.1 ± 44.8
3	−11.3	−12.4	815.9	−11.98 ± 0.3	**−12.4**	66.1 ± 27.9
4	−11.0	−13.0	1092.2	−12.10 ± 0.3	−12.4	65.1 ± 11.6
5	−11.0	−12.6	1515.7	−11.86 ± 0.9	**−12.8**	72.4 ± 25.9
6	−11.3	−12.9	2730.4	−12.08 ± 0.4	−12.7	74.8 ± 19.9
7	−11.3	−13.0	3546.9	−11.92 ± 0.9	−12.8	61.8 ± 49.9
8	−11.3	−11.4	3661.7	−10.80 ± 0.5	**−11.4**	44.3 ± 18.2

**Table 8 ijms-22-13226-t008:** New ligands with best binding energy using the heuristic search approach.

G	New Ligand (SMILES)
1	Clc1ccc(CN2CCNCC2)cc1
2	CN(NC1c2ccccc2-c2ccccc21)c1cnc2ccccc2c1
3	NCc1cc2c(NNC3c4ccccc4-c4ccccc43)c[nH]c2cc1O
4	Cc1cn(-c2ccnc3[nH]ccc23)c2c1C(=O)CC(C)(C)C2
5	Cc1cccc(-c2nc3cc(F)ccc3[nH]2)c1
6	CNc1ccc(-c2ccnc3[nH]ccc23)cc1-c1ccccc1C
7	CNc1cc(NC2c3ccccc3-c3ccccc32)c(C)c(N2CCCCC2)c1Cl
8	c1ccc(-c2ccccc2)cc1

## Data Availability

Datasets used in this research were downloaded free of charge from the ACS Publications website at DOI: 10.1021/acsmedchemlett.8b00397.

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
