# Peer review of "Development of Computational Approaches with a Fragment-Based Drug Design Strategy: In Silico Hsp90 Inhibitors Discovery"

_ijms, 2021, doi:10.3390/ijms222413226_

Round 1

Reviewer 1 Report

This manuscript carried out fragmentation and reconstruction of the molecule, and carried out the inhibitor search with the stronger binding affinity based on the structure of the inhibitor of Hsp90. In this calculation, it became possible to also carry out the lowering of the calculation cost by introducing the semi-exhaustive method. This method is based on the FBDD and LBDD methods, and it is considered to be useful as one of the rapid inhibitor search methods. Therefore, I would recommend it for acceptance after the minor points listed above and annotated on the manuscript are addressed.

Comments:

1) Though various techniques are known to construct molecules from fragments, it should be argued that your method is superior to those methods.

2) Need a little more variation in the functional groups used in the body to look for inhibitors with stronger binding affinities?

3) Lipinski rules is crucial in terms of medicinal products that can be oral administration compounds. However, from the viewpoint of obtaining stronger binding affinities, it is considered that more diverse compounds could be obtained without using Lipinski rules.

4) Are there constant correlations between Autodock scores and activity values? If there is almost no correlation, the evaluation by Autodock will be meaningless.

5) It is unclear by what criteria the eight groups were classified, so it should be explained in detail.

Minor points:

6) SMILE is mistake of SMILES. (Need S)

7) In Fig.6, compound G5 is wrong structure. SH(OH)2 is mistake of SO2.

Reviewer 2 Report

The manuscript by Leon et al. describes a fragment based drug design strategy applied to Hsp90 inhibitors discovery.

The topic seems interesting but the manuscript lacks in clarity, partially due to a low fluency in English language. Concerning this aspect, the manuscript has to be strongly revised (e.g. remove mixtures past/present, singular/plural, a/an, carefully check the construction of the phrase).

Here following my main critical observations and questions:

Introduction:

“In order to evaluate our method we have constructed a library of several Hsp90 ligands that has been already addressed [20–24], and several binding sites have been determined. However, the problem of finding acceptable ligands with a high inhibition level remains unsolved. Currently, a database of more than Hsp90 ligands [25] is available.

a) Which is the method you are evaluating? Until the previous lines only general considerations are reported. Describe the method.

b) the reference is to PDB. And PDB it is not a database of Hsp90 ligands. Cite the proper database where these ligands are collected or write the sentence in a different way.

c) what does it means “of more than Hsp90”?? It is evident we miss a number. Add it.

d) remove the zero and start the paragraph with number 1.

Materials and Methods

“The proposal is based on an FBDD approach. Particularly, the computations of new  ligands is based on a deconstruction-reconstruction approach (DRA). In Figure 1 the stages of the algorithm can be observed. Each stage is explained in the following:”

The paper is not a project. So do not use the word “proposal”. Computations is plural, “is” is singular…which of the two?

I suggest “each stage is explained as follows”.

“Stage 1. Deconstruction. A set of Hsp90 ligand” . Please define the number of the ligands. We should know how many.

“Stage 2. Duplicity and Cleaning. For each sub-ligand obtained in the first stage”. Please define the number of the sub-ligands obtained. How many?

Final consideration: you used smiles from the 3D structure. What about chiral centres? What about ionization state at pH 7.4? How did you manage these aspects? Add these details in the text.

Discussion

“We can assume that the variety of the group can influence in the binding energy improvement.” Please explain this sentence and re-write it in a clearer way.

“According to the new ligands obtained, we can observe that the group 4 and 6 obtained the same new ligand”. I do not agree. The ligands obtained with the first approach are ALL structurally similar, four over six (67%) present slight structural variations. Something is wrong in designing the method? Please add as supplementary material a table with all the structures you used in this study. How many? How similar? This is a very important information for evaluating this study.

Conclusions

This section has to be totally re-written in an acceptable English. In the present form it is impossible to understand Authors conclusions.

Last but very important comment: you should test the validity of your computational method biologically testing the best three compounds you have selected. Tests on Hsp90 are not very long or very expensive, thus you can add them to complete the manuscript.

Reviewer 3 Report

I found the paper interesting and the results are good for the general audience. However, I think the paper needs some minor modifications,

  1. The manuscript of the language must be improved in the revised version.
  2.  Limitations of the study must be highlighted in the discussion section. 
  3. What are the future recommendations, that must be highlighted in the conclusion section. 

Round 2

Reviewer 2 Report

The manuscript can be accepted in the present form, after a second English language revision.

Author Response

Thank you for your comments and suggestions. We checked the full document again and corrected all the errors we found.